# In Situ Surface-Initiated Atom-Transfer Radical Polymerization Utilizing the Nonvolatile Nature of Ionic Liquids: A First Attempt

**DOI:** 10.3390/polym13010061

**Published:** 2020-12-25

**Authors:** Ryo Satoh, Saika Honma, Hiroyuki Arafune, Ryo Shomura, Toshio Kamijo, Takashi Morinaga, Takaya Sato

**Affiliations:** Department of Creative Engineering, National Institute of Technology, Tsuruoka College, 104 Sawada, Inooka, Tsuruoka, Yamagata 997-8511, Japan; r-satoh@tsuruoka-nct.ac.jp (R.S.); saika@tsuruoka-nct.ac.jp (S.H.); harafune@tsuruoka-nct.ac.jp (H.A.); shomura@tsuruoka-nct.ac.jp (R.S.); kamijo@tsuruoka-nct.ac.jp (T.K.); takayasa@tsuruoka-nct.ac.jp (T.S.)

**Keywords:** in situ polymerization, ionic liquid-type polymer brush, surface-initiated atom-transfer radical polymerization

## Abstract

In this paper, in situ surface-initiated atom-transfer radical polymerization (SI-ATRP) based on both an open and a coated system, without using volatile reagents, was developed to overcome the limited usage of ATRP due to the necessity of sealing. Nonvolatile ionic liquid (IL)-type components were used, specifically *N*,*N*-diethyl-*N*-(2-methacryloylethyl)-*N*-methylammonium bis(trifluoromethylsulfonyl)imide as the polymerizable monomer and *N,N*-diethylmethyl(2-methoxyethyl)ammonium bis(trifluoromethylsulfonyl)imide as the polymerization solvent. In the experiment, the reversible-deactivation radical polymerization characteristics are properly ensured in nonvolatile ATRP solution coated on silicon wafer as thin liquid film, to form concentrated polymer brushes (CPBs). The average molecular weight and molecular-weight distribution of the polymer produced in the liquid film and formed on silicon wafer were measured by gel permeation chromatography, which confirms that the polymerization reaction occurred as designed. Furthermore, it is clarified that the surface of the polymer brush synthesized in situ swollen by IL also exhibited low friction characteristics, comparable to that synthesized in a typical immersion process. This paper is the first to establish the effectiveness of in situ preparation for CPBs by using the coating technique.

## 1. Introduction

In 1998, Tsujii and coworkers developed surface-initiated atom-transfer radical polymerization (SI-ATRP) to synthesize concentrated polymer brushes (CPBs) that have a brush density per area that is more than ten times larger than traditional semi-diluted polymer brushes [1]. CPBs are highly swollen and stretched in a polymer brush subjected to a good solvent, and they have special surface properties collectively known as the CPB effect; these include a high resilience, a size exclusion effect [2], and a low frictional property of the polymer brush surface, which is derived from the high osmotic pressure of the CPB layer [3]. Although CPBs are characterized by soft material, their surface has an extremely low coefficient of friction (CoF) and can withstand an applied pressure of more than 400 MPa, as demonstrated by our research group [4]. In light of these advantageous characteristics, a new academic and technical field has been established in Japan: the study of soft and resilient tribology material [5]. Based on the earlier findings, many companies are developing low-frictional soft materials to be used in sliding mechanical parts, to avoid damaging the facing sliding surface [6].

The authors developed an ionic liquid (IL) with a polymerizable double-bond functional group on the cation of the IL [7], to create a nonvolatile and nonflammable polymer solid electrolyte for electrochemical devices [8]. The IL-type polymer materials exhibit extremely high ionic conductivity [9] and a high mechanical strength under severe conditions, such as high vacuum or long exposure to high temperatures [10]. CPBs swollen with an IL with a lubricating function create a soft tribology system, which can be used under severe conditions [11].

Unfortunately, there are several problems with respect to the practical application of CPBs by SI-ATRP, using volatile monomers and/or solvents. ATRP maintains its reversible-deactivation radical characteristics based on chemical equilibrium [12], and, therefore, small changes in chemical composition due to volatilization of monomer and/or solvent can result in systematic deviations from the designed molecular weight and molecular weight distribution. Moreover, as the molecular weight distribution increases, the CPB density per unit area decreases, and, as a result, the CPB effect deteriorates [13]. To avoid the volatilization, SI-ATRP has been performed in a batch process, using a sufficient amount of solvent to ensure the entire surface is submerged. Unfortunately, addressing these limitations will increase production costs. Thus, we focused on utilizing coating technology that is advantageous for imparting the required CPB with respect to the required surface. Furthermore, the processes of coating liquid film of a polymerization solution onto an object in an open system and performing polymerization are clearly eco-advantageous over the process of immersing [14].

For the application of coating technology to SI-ATRP, the polymerization solution must be completely nonvolatile. Technology using nonvolatile IL as the polymerization solvent is well established [15], but for complete SI-ATRP in liquid film polymerization, even monomers must not volatilize. Until now, the aim of researchers who have adopted ILs as polymerization solvents has been limited to examine the polymerization rate or polydispersity of obtained polymers in IL [16]. Therefore, as far as the authors are aware, a radical polymerization system focusing on the nonvolatile nature of the IL is yet to be documented, in which both the monomer and the polymerization solvent are nonvolatile. Therefore, in this study, an in situ SI-ATRP polymerization system was designed with nonvolatile components, using the polymerizable IL *N*,*N*-diethyl-*N*-(2-methacryloylethyl)-*N*-methylammonium bis(trifluoromethylsulfonyl)imide (DEMM–TFSI) as a monomer and *N,N*-diethyl-*N*-methyl(2-methoxyethyl)ammonium bis(trifluoromethylsulfonyl)imide (DEME–TFSI) as the polymerization solvent. The process involved the coating of a polymerization solution on a silicon substrate, on which an ATRP-initiating group is immobilized, and then heating the substrate in an open system to perform SI-ATRP. Our main interest was whether the reversible-deactivation radical polymerization characteristics in an open system using ILs are properly ensured. To verify this process, the average molecular weight and molecular-weight distribution of the polymer produced in the liquid film were investigated in chronological order. Furthermore, by cutting out the polymer brush generated on the silicon substrate, using hydrofluoric acid (HF) [17,18], and measuring the average molecular weight thereof, we confirmed that the polymerization reaction occurred as designed. This paper is the first to establish in situ preparation for CPBs by using the coating technique (Scheme 1).

## 2. Materials and Methods

### 2.1. Materials

Ethyl 2-bromoisobutyrate (2-(EiB)Br, 98%) and 2,2′-bipyridine (Bipy, 97%) were used as received from Nacalai Tesque Inc. (Osaka, Japan). Copper(I) chloride (Cu(I)Cl, 99.9%) was purchased from Wako Pure Chemicals (Osaka, Japan). Copper(II) chloride (Cu(II)Cl_2_, 98%) was obtained from Nacalai Tesque Inc. (Kyoto, Japan). DEME–TFSI and HF (50% aqueous solution) were purchased from Kanto Chemical Co. Inc. (Tokyo, Japan), and DEMM–TFSI was synthesized according to the methods previously reported [7]. DEMM–TFSI was purified by passing through a column of activated basic alumina to remove the inhibitor. All other reagents were used as received from commercial sources.

### 2.2. In Situ Preparation of CPBs of Poly(DEMM–TFSI) by SI-ATRP

The ATRP of DEMM–TFSI was performed as follows: In a typical run, a flask was charged with Bipy (26.0 mg, 0.13 mmol), DEME–TFSI (1.96 g), and acetonitrile (8.0 g), and the mixture was deoxygenated by purging with argon for 10 min. In a glove box purged with argon, Cu(I)Cl (5.2 mg, 0.052 mmol) and Cu(II)Cl_2_ (0.78 mg, 0.0058 mmol) were added to the mixture, and a three-way stopcock was attached to the flask. Then, while closing the stopcock and maintaining the sealed state, the flask was taken out of the glove box and stirred in an oil bath at 60 °C, to completely dissolve the copper and Bipy. Then, the flask was transferred to the glove box, and the solution was filtered with a membrane filter (pore size of 0.2 μm). Then, acetonitrile was removed by evaporation, using a vacuum pump. The reason for adding this sacrificial acetonitrile is to reduce the viscosity of the solution and promote the dissolution of copper and ligand. To suppress the precipitation of copper chloride during the polymerization process, the reaction was performed with a 10% excess of the ligand (Bipy) for the required molar amount of copper. Then, another Schlenk tube was charged with DEMM–TFSI (8.0 g, 16.6 mmol) and 2-(EiB)Br (0.65 mg, 0.0033 mmol), and the mixture was deoxygenated by purging with argon for 20 min. A nonvolatile ATRP polymerization solution was prepared by mixing the aforementioned solutions in the Schlenk tube and the flask.

Because the ATRP polymerization solution prepared with only nonvolatile components had a higher viscosity than the conventional one, it was maintained as a liquid layer with a thin liquid film on a silicon substrate until polymerization was complete. If the undissolved solid component of copper chloride (or bubbles) mixed in during solution preparation adheres to the silicon substrate, a polymer brush cannot be formed on that portion. Therefore, to produce a uniform polymer brush layer, it was essential to filter the copper complex solution and de-foam the polymerization solution by centrifugation. Using a centrifuge installed in the glove box, the polymerization solution was de-foamed by centrifuging at 500× *g* for 10 min.

A polymerization solution was coated on a silicon wafer (area of 10 cm^2^) with the fixed initiator according to the previous report [5]. The same treatment was applied to six silicon substrates for the time series analysis. Apart from the use of excess ligand, the stoichiometry of the reaction system was [DEMM–TFSI]_0_/[2-(EiB)Br]_0_/[Cu(I)Cl]_0_/[Cu(II)Cl_2_]_0_/[Bipy]_0_ = 5000/1/20/5/50. The polymerization was performed in an incubator (installed in the glove box) at 70 °C, and, after a prescribed time, an aliquot of the coated solution was taken out to obtain the following: nuclear magnetic resonance (NMR) measurement to estimate monomer conversion and gel permeation chromatography (GPC) measurement to determine the molecular weight and the distribution of the free chains produced in solution from the free initiator 2-(EiB)Br.

The polymer-grafted silicon wafer was immersed in an aqueous acetonitrile solution (50/50 *v*/*v* mixture) containing 10% HF for 3 h to obtain cutoff sample of CPBs. The solution containing the graft polymer that was cleaved and eluted in the solution was neutralized with a saturated aqueous solution of sodium hydrogen carbonate; it was then transferred to a dialysis tube and dialyzed with an excess amount of GPC solvent (50/50 *v*/*v* mixture of acetonitrile and water containing 0.5 M acetic acid and 0.2 M NaNO_3_).

### 2.3. Characterization of Poly(DEMM–TFSI) Obtained by In Situ SI-ATRP

^1^H NMR (400 MHz) spectra were measured for a CD_3_CN solution of samples and reported in ppm (δ) from the residual solvent peak (δ = 1.94), using a JEM-ECX400 spectrometer (JEOL Ltd., Tokyo, Japan). GPC was performed on a Shodex GPC-101 high-speed liquid chromatography system equipped with a guard column (Shodex GPC KF-G, Showa Denko K.K., Tokyo, Japan), two 30 cm mixed columns (Shodex GPC KF-806L, exclusion limit = 2 × 10^7^), and differential refractometer (Shodex RI-101) and multi-angle laser light scattering (MALLS) detectors (Wyatt Technology DAWN8^+^, Wyatt Technology, Santa Barbara, CA, USA); acetonitrile/water (50/50 *v*/*v* mixture), 0.5 M acetic acid, and 0.2 M NaNO_3_ were used as eluents. An IRAffinity-1 Fourier transform infrared spectroscopy (FTIR) instrument (Shimadzu Corporation, Kyoto, Japan) equipped with a single-reflection MIRacleA attenuated total reflectance (ATR) accessory (Shimadzu Corporation) with a germanium prism. The following parameters were given: measurement range of 4000–750 cm^−1^, acquisition of 16 scans, and resolution of 4 cm^−1^. The data obtained with the FTIR were processed by IRsolution (Shimadzu Corporation, version 1.60). A JSM-7100F scanning electron microscope (SEM; JEOL Ltd.) was used to observe the cross-section of the grafted polymer brush layer. A JED-2300F energy-dispersive X-ray spectroscopy (EDX) equipment (JEOL Ltd.) was used for the elemental analysis of the cross-section. The experimental parameters for SEM and EDX consisted of the following: acceleration voltage of 7.5 and 10 kV for SEM and EDX, respectively, current of ten arbitrary units (actual value of 7.475 nA), and detection signal for SEM secondary electrons.

### 2.4. Friction Force Measurements

Lubrication properties of poly(DEMM–TFSI) formed by coating or immersion process were evaluated by macroscopic friction measurements. We applied ball-on-plate-type reciprocating tribometer (Tribogear type-38, Shinto Scientific Co. Ltd., Tokyo, Japan). A poly(DEMM–TFSI) substrate was fixed on the sliding stage, and a glass ball was set in the ball holder connected to a load cell. We measured the friction forces between the glass ball and poly(DEMM–TFSI) by sliding the sample stage over a range of 10 mm, at various sliding speeds, under a normal load of 0.98 N, to evaluate the lubrication regime. *N*-(2-methoxyethyl)-*N*-methylpyrrolidinium bis(trifluoromethanesulfonyl)imide (MEMP-TFSI) was dropped on the surface of sample substrates as a nonvolatile liquid lubricant.

## 3. Results and Discussion

### 3.1. In Situ SI-ATRP of Polymerizable IL on a Silicon Wafer

A nonvolatile ATRP solution was coated on a substrate, on which starting groups were immobilized, and the synthesis of CPBs under in situ conditions in an open system was investigated. As shown in the experimental section, undissolved solid components of copper chloride and bubbles were carefully removed by filtering and centrifugation, to avoid the defect of CPB layer on silicon substrate.

Under the current polymerization conditions, the final degree of polymerization is 5000 (~2.3 *×* 10^6^ when converted to molecular weight). Therefore, the calibration curve of general GPC analysis using a polyethylene glycol standard sample could not cover a wide molecular weight range, and thus, it was difficult to calculate an accurate number average molecular weight and molecular weight distribution. In this study, we used the absolute molecular weight calibration curve that we prepared by using GPC–MALLS analysis of the standard sample with the *dn*/*dc* value of poly(DEMM–TFSI) in GPC solvent set as 0.197, which has been previously reported by the authors [9].

The polymerization characteristics were evaluated by analyzing the free polymer produced in the ATRP polymerization solution coated on the silicon wafer and the graft polymer cleaved from the surface of the silicon wafer by HF treatment. In the ATRP experiment, the monomer conversion reached 99% after 70 h, which produced a polymer with a number average molecular weight of 1,826,000 and a dispersity (*M*_w_/*M*_n_) of 1.20. Figure 1 shows the variation in ln([*M*]_0_/[*M*]) versus polymerization time for the polymerization of DEMM–TFSI in DEME–TFSI at 70 °C with Cu(I)Cl/Cu(II)Cl_2_/Bipy as a catalyst. Almost full conversion was reached after 70 h, where an almost linear first-order kinetic plot passing through the origin can be observed until almost 100% conversion.

Figure 2 shows the evolution of the number average molecular weight (*M*_n_) and the *M*_w_/*M*_n_ of the cleaved graft polymer and the free polymer produced from the free initiator, from which it is evident that the *M*_n_ of the graft and of the free polymers are nearly the same, both of which increased in proportion to monomer conversion.

The values of the *M*_n_ estimated by GPC–MALLS deviated from the theoretical values calculated with the initial molar ratio of DEMM–TFSI to the free initiator (dotted line), as shown in Figure 2. In a previous study, we observed a good agreement of the theoretical molecular weight and the experimental molecular weight measured by GPC–MALLS, where the molar ratio of DEMM–TFSI to the ATRP initiator was 500/1 [4]. Therefore, the obtained deviation can be attributed due to a partial termination of the free initiator of the process of reaching the stationary state of the redox equilibrium of the copper chloride complex at the initial stage of polymerization. In this experiment, the amount of the free initiator (3.3 μmol) is so small compared with a previous report (113 μmol) that deactivation due to termination could not be ignored. In other words, when the molar ratio of DEMM–TFSI to the ATRP initiator was 500/1, the amount of the terminated initiator was negligible, resulting in a good agreement of the theoretical and the experimental molecular weight. As a result, the molar ratio changed from the initial value ([DEMM–TFSI]_0_/[2-(EiB)Br]_0_ = 5000/1), and, as such, the free and grafted number average molecular weight values deviated from the theoretical values. The actual molar ratio of DEMM–TFSI to the available ATRP initiator was estimated to be 9000/1 according to experimental results, described as a broken line in Figure 2.

As shown in Figure 2, the dispersity remains lower than 1.3 for most samples. These results confirm that the reversible-deactivation nature of polymerization for the DEMM–TFSI initiated from the silicon wafer and that the silicon wafer is grafted with well-defined poly(DEMM–TFSI). It should be emphasized that this is the first successful in situ synthesis of CPBs on a solid surface in an open coating reaction system by SI-ATRP.

### 3.2. Characterization of Poly(DEMM–TFSI) on a Silicon Wafer by ATR/FTIR

To identify the brushes on the silicon wafer, ATR/FTIR was performed. Figure 3 shows the ATR/FTIR spectrum of poly (DEMM–TFSI) on the silicon wafer. The grafted side of the silicon wafer was allowed to give the ATR prism directly. The peak at the wavenumbers 1730, 1348, 1182 (with a shoulder 1227), 1134, and 1053 cm^−1^ were assigned to C=O stretching from the main chain of methacrylates (ester bond), S=O stretching, C–F absorption, and S–N–S asymmetric stretching, respectively. The results were reasonable for the previous study [7,19]. The broad peak of 1485–1462 cm^−1^ was assigned to the C–H bending of the methyl group in the DEMM structure [20]. Disappearance of C=C stretch absorption band at 1638 cm^−1^ derived from the methacrylate group can be attributed to the fact that the C=C double bond was consumed when the monomer DEMM was polymerized. No other peaks can be observed from 2000 to 4000 cm^−1^. Hence, the ATR method confirmed the existence of poly(ILs) on the silicon wafer.

### 3.3. Cross-Sectional Observation of the Poly(DEMM–TFSI) Grafted on the Silicon Wafer and Elemental Analysis Using SEM and EDX

Direct observation of the grafted CPBs was attempted by using SEM; the visualization of layered CPB distribution was executed by EDX analysis (Figure 4). The samples were from the same lot as the one subjected to ATR/FTIR measurements. Figure 4a shows the SEM image of a sample cross-section. The trace of cleavages imaged out the pure silicon crystal of the wafer in the lower region of the image. In the middle part, the layered structure is separated from the lower region. EDX analysis was performed to identify the brush layer on the silicon wafer. Extracted EDX mapping images (Figure 4b, red; Figure 4c, green) indicate the distributions of carbon and silicon, respectively. The carbon is representative evidence of the main chain element of poly(DEMM–TFSI) in Figure 4b; the silicon component was seen to locate under the grafted layer in Figure 4c. The over-layered images of Figure 4b,c indicate that CPBs were successfully grafted on the surface of the silicon wafer, thereby confirming that in situ SI-ATRP on the silicon wafer in an open coating reaction system took place. The EDX spectrum in Figure 4e shows the elemental composition on the surface layer of the polymer. The electron beam in the SEM–EDX equipment was irradiated vertically onto the sample surface. In the spectrum, carbon, nitrogen, oxygen, fluorine, silicon, and sulfur were identified, but the silicon was from the silicon wafer itself. The origin of each element can be assumed as follows: carbon, which is the main component of the organic poly(ILs), and oxygen come from the poly(DEMM) backbone and TFSI; nitrogen and fluorine come from the TFSI. No other peaks can be observed above 3 keV. Hence, poly (ILs) grafted onto the silicon wafer by SI-ATRP could be easily allowed to perform qualitative analysis, and the demonstrated in situ polymerized CPB layer was chemically determined by SEM–EDX. Since heat shrinkage of the polymer brush layer occurs due to electron beam irradiation, it is difficult to obtain an accurate film thickness from the SEM image. Therefore, the graft density and surface occupancy were calculated according to the film thickness measured by ellipsometry. The dry film thickness measured by ellipsometry was 515 nm, the graft density was calculated to be 0.15 chains/nm^2^, and the surface occupancy was calculated to be 35%.

### 3.4. Lubrication Properties Using IL Polymer Brushes Formed by In Situ Synthesis

We examined the influence of the fabrication process of CPBs (coating or immersing) on the sliding speed dependence of the CoF by measuring friction at the glass ball/poly(DEMM–TFSI) interface under the normal load of 0.98 N (Figure 5). The CoF of glass ball/poly(DEMM–TFSI) formed by the coating process showed a low value (10^−3^ order) and decreased with the decrease of a sliding speed. This result suggests that it was in hydrodynamic lubrication regime, where the glass ball and CPB layer were separated by viscous liquid layer of lubricant, avoiding direct contact of glass and substrate of CPBs. These results are consistent with our previous studies. We found that the surface roughness of the counter face of brush layer was important to preserve low friction of thin CPB substrate [5]. Because maximum height difference of conventional glass ball (220 nm) was much higher than the thickness of brush layer (40 nm), a rough glass ball was not compatible with avoiding brush wear caused by solid contact of the glass and brush substrate. Increasing thickness of CPB layer was effective to maintain low friction against the hard steel sphere under a high Hertzian contact pressure (540 MPa) [11], to result in viscous-type friction. Thus, the obtained results can be explained by dry thickness of 515 nm being thick enough to avoid solid contact of the glass ball and brush substrate, resulting in low friction. Additionally, because the CoF values of these two fabrication processes are comparable, it is evident that poly(DEMM–TFSI) formed by the in situ process can equally preserve the lubrication properties, when compared with typical immersion process. These results strongly support the idea that an in situ process of CPBs utilizing ILs can provide a cost-effective and ecofriendly way to form resilient functional coating, which can be utilized for fabricating low frictional materials such as seal or bearings with high energy efficiency and a long lifetime.

### 3.5. Future Perspective

Although SI-ATRP technology is now spreading worldwide [15], in situ SI-ATRP with a reusable and oxygen-durable system is required for industrial process. Activators regenerated by electron transfer, in combination with ATRP with a mechanism to reduce the oxidized transition metal complexes, have been proposed [21] as a solution to oxygen contamination; however, the need for a closed system to prevent solvent volatilization is yet to be realized.

Hence, the authors plan to further investigate IL solvent systems containing catalysts, to achieve repeatable reversible-deactivation radical polymerization with optimized monomer concentration and solvent viscosity during polymerization. In the future, we aim to develop an in situ coating of SI-ATRP that features reuse of the reaction solution under oxygen-durable and open conditions for industrial applications of CPB materials.

## 4. Conclusions

In this study, SI-ATRP in both open and coated forms, using nonvolatile DEME–TFSI as the solvent, was achieved for the first time in a complete IL system, with DEMM–TFSI as the monomer. As the conversion progressed, the number average molecular weight was sufficiently regressed in a linear fashion. The evaluation of the number average molecular weight and the dispersity of the graft and free polymers as a function of monomer conversion for the polymerization of DEMM–TFSI with an initiator-coated silicon wafer were confirmed. Simultaneously, the molecular weight distribution remained below 1.3, thereby proving the reversible-deactivation nature of the polymerization. To determine the brush structure, ATR/FTIR measurements were performed, and the molecular vibrations were attributed to the DEMM main chain and the TFSI anion. SEM–EDX observations show that the SI-ATRP was achieved in the reaction system, and the layered structure of poly(DEMM–TFSI) was analyzed microscopically. Each element derived from poly(DEMM–TFSI), which mainly consists of carbon, nitrogen, oxygen, fluorine, and sulfur, was identified chemically. Consequently, the concept of SI-ATRP polymerization in open systems using nonvolatile reagents was demonstrated.

## Data Availability

This paper contains all the data used in the discussion.

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
