# Peer review of "In Situ Surface-Initiated Atom-Transfer Radical Polymerization Utilizing the Nonvolatile Nature of Ionic Liquids: A First Attempt"

_polymers, 2020, doi:10.3390/polym13010061_

Round 1

Reviewer 1 Report

This paper deals with the investigation of surface-initiated atom transfer radical polymerization (ATRP) N,N-diethyl-N-(2-methacryloylethyl)-N-methylammonium bis(trifluoromethylsulfonyl)imide (DEME–TFSI) using corresponding ionic liquid as a non-volatile solvent to prepare concentrated polymer brushes. The main feature of this article is developing of the simple procedure to controlled grafting of corresponding ionic liquid monomer to silicon wafer in open system using non-volatile reagents. The living nature of the polymerization of DEME–TFSI as well as successful grafting of this monomer into silicon wafer were confirmed by different analytical techniques. I recommend accepting this paper for publication in Polymers after following minor revisions:

  1. I suggest authors adding Scheme describing the polymerization process developed in this work.
  2. How can authors explain some difference between Mn (free) and Mn (graft) (Figure 2)?
  3. Does the lover efficiency of initiation observed at such high [M]/[I] ratio as 5000 influence the density of prepared brushes?
  4. Page 7, line 251: change Figure 5c to Figure 4c.

Reviewer 2 Report

I very rarely do this but I have to state that this is an interesting paper that will really be of interest to the readers of Polymers. This is a far over average quality article that describes new and highly interesting data on a very relevant subject: new polymeriztion tools for a wide range of surface applications.

Overall, the article is very straightforward and well written. The data are clear and well presented and I clearly recommend acceptance.

the authors should do a spell check as I have seen a number of typos such as "frask" instead of flask in line 97 and line 106.

Then I have two questions and a suggestion:

  1. Why did the authors not try to perform the reaction without degassing? The solubility of many gases in ILs is not that high (not sure about oxygen in th eILs use here) and it may thus be that the reaction still proceeds even in the presence of some oxygen due to the high stability of the radicals in ILs, which is well known.
  2. Is there evidence of Cu left in the samples after washing? The EDX pectra only cover the range up to 3 keV, which would not show Cu.
  3. Suggestion: there are Cu(II) based ILs out there, so it may be interesting to evaluate the reaction in these ILs as well. Authors that have published such ILs include Duncan Bruce and Kenn Seddon (who first reported them in 1996), Francisco Neve, Andreas Taubert, Peter Nockemann, John Holbrey and a number of others.

Overall this is a highly relevant article and I clearly recommend acceptance.
